# The Influence of Industrial Output, Financial Development, and Renewable and Non-Renewable Energy on Environmental Degradation in Newly Industrialized Countries

Shabana Parveen [1], Saleem Khan [2], Muhammad Abdul Kamal [2], Muhammad Ali Abbas [1], Aamir Aijaz Syed [3] and Simon Grima [4,5,*]

[1] Department of Economics, Hazara University (KP), Dhodial, Mansehra 21120, Khyber Pakhtunkhwa, Pakistan
[2] Department of Economics, Faculty of Business and Economics, Abdul Wali Khan University Mardan (KP), Mardan 23200, Khyber Pakhtunkhwa, Pakistan
[3] Institute of Management, Commerce, and Economics, Shri Ramswaroop Memorial University, Lucknow 225003, India
[4] Department of Insurance and Risk Management, Faculty of Economics, Management and Accountancy, University of Malta, MSD 2080 Msida, Malta
[5] Faculty of Business, Management and Economics, University of Latvia, 1586 Riga, Latvia
* Correspondence: simon.grima@um.edu.mt

**Abstract:** The prime objective of this study is to examine the impact of industrial output and financial development on carbon dioxide emissions for a panel of 10 newly industrialized countries, namely Brazil, China, India, Indonesia, Malaysia, Mexico, Philippines, South Africa, Thailand, and Turkey. The empirical analysis was conducted between 1982 and 2019 by employing various estimation tests and techniques. The different tests account for cross-sectional dependence in different series of the model. Therefore, the relevant panel unit root was conducted, and we found that all series become stationary after the first difference. The long run parameters were estimated, and we found that there is a significant long-run relationship between the industrial output, the financial development, and the carbon emissions. The carbon emissions are found to be significantly affected by both domestic income and industrial output, while being negatively affected by financial development. Industrial production coefficient estimates are highly elastic when compared to the other estimates. The results also indicate unidirectional short-run causality from the domestic output and trade openness to carbon emissions, urban population to domestic output, and financial development to industrial output. However, there is no evidence of bidirectional causality. The study concludes that sustainable economic growth can be achieved by using contemporary and efficient production techniques, using environmentally friendly inputs in industries, and increasing vigilance of both the public and private sectors. Both the public and private sectors should therefore be pushed to use more modern, eco-friendly, and productive processing techniques. It is recommended that both the public and commercial sectors be encouraged to embrace cutting-edge, environmentally friendly, and productive processing methods.

**Keywords:** financial development; industrial output; carbon emissions; newly industrialized countries; renewable energy; environmental degradation

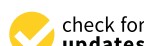



## 1. Introduction

Global warming has become a challenge that the world is facing currently. The major factor behind global warming is $CO_2$ emissions that weaken the ecological state and lead to environmental degradation. On the other hand, energy is a prerequisite for economic activities today. In broad terms, a substantial amount of energy is used by four sectors: the transport sector, commercial sector, residential sector, and industrial sector. It is assumed that the industrial sector consumes 51% of the world's energy. A sector's success leads

to further energy consumption, which contributes to a rise in fossil fuels that have an adverse environmental impact, resulting in increased concern about the environmental consequences of industrial production [1]. In recent times, increased global focus has been presumed to curtail carbon emissions from energy production and industrial activities. The Kyoto Protocol (1997) and the Paris Agreement (2015) are fundamental steps towards reducing emissions of greenhouse gases (GHGs) to sustain a global temperature rise below 2 °C above pre-industrial levels and to explore the necessary steps further to restrict the increase in temperature to 1.5 °C.

The newly industrialized countries (NIC) contributed 42% to total global $CO_2$ emissions due to acceleration in their economic growth [2]. As time passed, the structure of the countries' economies shifted from agricultural to industrial or service-based economies. To attain high growth rates or compete with developed countries, new industrial countries are expanding their industrial sectors to produce more goods while disregarding pollution standards [3]. As a result, air pollution and ecosystem degradation are worse in newly industrialized countries (NIC) than in other countries [4]. $CO_2$ is a part of the atmosphere but has increased dramatically by one-third after the industrial revolution [2]. NASA data (2016) [5] show that atmospheric $CO_2$ remained below 300 parts per million (ppm) for 650,000 years but is now at 400 ppm. It depicted that $CO_2$ emissions grew with the industrial revolution. More energy use in industries is a key factor behind high $CO_2$ emissions. More energy use in sectors of the economy could be a major factor behind high $CO_2$ emissions, although it is not the only factor. In addition, financial development is an important determinant of $CO_2$ emissions. Financial development can, in many ways, impact $CO_2$ emissions. First, financial development improves stock market conditions, and listed companies can increase their monetary investment in new projects owing to financial growth. This feat raises installations and services that need more energy and result in high $CO_2$ emissions. Second, financial development motivates foreign direct investment in a country that promotes economic growth and aggravates $CO_2$ emissions. Third, an efficient financial development program positively affects purchasing expensive consumer goods such as bigger houses, air conditioners, automobiles, and other products. All of these considerations contribute to energy consumption and high $CO_2$ emissions.

The spike in individual activities triggers an increase in carbon dioxide ($CO_2$) emissions due to the acceleration of conventional energy sources. By 2025, the primary energy use in emerging economies is projected to reach an average of 3.2% per year [3,4]. In line with these realities, the International Energy Outlook (IEO) foresees a rapid rise in global access to low energy by 2025. This figure forecasts the terrifying situation of the contribution of traditional global energy demand, primarily from fossil fuels, which is between 80 and 95% [6]. It is, therefore, essential to incorporate modern and sustainable sources of energy to regulate pollution without harming economic development and the climate. For these purposes, economies worldwide strive to use modern renewable energy sources, such as biomass, tidal, wind, geothermal, solar, and hydroelectric power. Gielen et al. [7] pointed out that by 2050, renewable energy resources will reach two-thirds of the world's overall energy demand. This will drastically reduce greenhouse gas emissions.

The major objective of this research work is to empirically analyze the causal nexus between carbon emissions, GDP growth, industrial output, financial development, and renewable and non-renewable energy consumption from 1982 to 2019 for 10 NICs. This study contributes to the existing body of literature in many ways. First, this study analyzes the impact of industrial output and financial development in NICs. To the best of our knowledge, only a few studies have examined the causal nexus among carbon emissions and variables like GDP, energy consumption, trade openness, and urbanization in the case of NICs. However, industrial output and financial development have not garnered much attention from researchers. The variables included in this study overcome the omitted variable bias problem. Second, previous research has concentrated on the effects of energy use while ignoring the structure of energy usage (i.e., renewable and conventional energy). As a result, we include both types of energy use in this study, which can provide

essential information for successful energy policies that contribute to achieving sustainable development goals. Lastly, we applied advanced econometric techniques, i.e., cross-section dependency statistics and panel cointegration tests, to make our analysis economically appropriate. We used a panel data method for long-run regression, i.e., FMOLS and the Granger causality test are used to know the nature of causality between the variables of interest. From the standpoint of policy, these empirical procedures are essential for developing alternatives for policies that might encourage cleaner and more sustainable consumption and production patterns in NICs [8].

The rest of the paper unfolds as follows. Section 2 provides a review of the literature. Section 3 describes the model specification, econometric technique, and data and description. Section 4 explains the empirical analysis and discussion, and Section 5 concludes the study with policy implications.

## 2. Review of Literature

A brief overview of previous research on the relationships between financial development, industrialization, green energy, and carbon emissions is provided in the following section.

### 2.1. Financial Development–Pollution Nexus

A plethora of studies empirically analyzed the role of financial development in affecting carbon emissions. For instance, many researchers (Acheampong, [9]; Adams & Klobodu, [10]; Al-Mulali, Ozturk, & Lean, [11]; Ganda, [12]; Nasir, Duc Huynh, & Xuan Tram, [13]; Shahbaz et al., [14]; Shoaib et al., [15]) have confirmed a positive impact of financial development on $CO_2$ emissions. In a similar vein, Y.-J. Zhang [16] studied the impact of financial development on carbon emissions for the Chinese economy over three decades. Based on the cointegration test and the variance decomposition method, the study found a positive and significant effect of financial development on carbon emissions. Likewise, Sadorsky [17] used panel data from 22 developing countries to analyze the nexus of financial development, energy consumption, and carbon emissions. The result shows that for these countries, financial development leads to more energy use that produces more carbon emissions. On the contrary, a growing body of literature has agreed that financial development reduces environmental degradation via the promotion of economic growth (e.g., Al-mulali, Tang, & Ozturk, [18]; Odhiambo, [19]; Shahbaz, Nasir, & Roubaud, [20]; Xing et al. [21]; Zafar, Saud, & Hou, [22]; Zaidi et al. [23], etc.). Besides, financial development promotes investment in the energy sector that assists in mitigating energy emissions [24]. Moreover, Nasreen and Anwar [25] reported a unidirectional causality from financial development to $CO_2$ emissions, whereas Shahbaz et al. [14] confirmed a bidirectional causality between the two variables. Boutabba [26] studied the Indian economy to identify the link between financial developments and carbon emissions. The results of the Johansen cointegration test and Error Correction Model (ECM) advocate that financial development contributes to environmental degradation. In addition, the Granger causality test identified one-way causality from financial development to carbon emissions. Ozturk and Acaravci [27] confirmed the long-run association between per capita carbon emissions, per capita real income, the square of per capita real income, and financial development in Turkey. The study revealed that a high level of per capita real income results in declining carbon emissions per capita. However, no significant impact of financial development on carbon emissions was found in the long term. The result further supported one-way causality from financial development to energy consumption in the short term.

Table 1 below provides a Chronological summary of the literature on financial development and $CO_2$ emissions.

**Table 1.** Chronological summary of the literature on financial development and $CO_2$ emissions.

| Authors | Time Period | Region/Countries | Findings |
| --- | --- | --- | --- |
| Abid [28] | 1996–2010 | 25 SSA countries | FD does not affect carbon emission |
| Abbasi and Riaz [29] | 1971–2011 | Pakistan | The effect of FD is insignificant on carbon emissions |
| Dogan and Turkekul [30] | 1960–2010 | USA | FD exerts no effect on $CO_2$ emissions |
| Saidi and Mbarek [24] | 1990–2013 | 19 emerging countries | FD reduces $CO_2$ emissions |
| Shahbaz, Nasir, and Roubaud [20] | 1955–2016 | France | FD has a declining effect on carbon emissions |
| Paramati, Mo, and Gupta [31] | 1991–2012 | G-20 countries | FD increases environmental quality by reducing $CO_2$ emissions |
| Adams and Klobodu [10] | 1985–2011 | 26 African countries | FD causes $CO_2$ emissions |
| Ganda [12] | 2001–2012 | OECD countries | FD increases $CO_2$ emissions |
| Charfeddine and Kahia [32] | 1990–2015 | MENA region | Inverted U-shaped relationship exit between FD and $CO_2$ emissions |
| Gokmenoglu and Sadeghieh [33] | 1960–2011 | Turkey | FD enhances carbon emissions |
| Shahbaz, et al. [14] | 1975–2014 | UAE | FD increases $CO_2$ emissions, and the relationship between financial development and $CO_2$ emissions is U-shaped and inverted N-shaped |
| Odhiambo [19] | 2004–2014 | 39 SSA countries | FD unconditionally reduces $CO_2$ emissions |
| Shoaib et al. [15] | 1999–2013 | G8 and D8 countries | FD has a significant and positive impact on carbon emission in both panels in the long-run |
| Ahmad et al. [34] | 1990–2017 | 90 Belt and Road countries | FD deteriorates the environmental quality by increasing the $CO_2$ emissions, and bidirectional causality exists between FD and $CO_2$ emissions |
| Ling et al. [35] | 1980–2017 | China | FD has important ramifications for carbon emissions |
| Shobande et al. [36] | 1995–2018 | OECD countries | FD reduces carbon emissions but increases energy use |
| Khan et al. [37] | 1984–2018 | 15 emerging and growth-leading economies | FD development across all dimensions reduces environmental quality ($CO_2$ emissions) |

Source: Authors' compilation.

### 2.2. Industrialization–Pollution Nexus

Some of the studies explored the industrialization impact on carbon emissions. Lin et al. [38] worked on data from 53 countries with different income levels, while Ma, Yan, and Cai [39], in the case of China, confirmed a significant association among financial development, industrialization, and $CO_2$ emissions. Wang, Shackman, and Liu [40] confirmed an association for Western developing countries; Abou-Ali, Abdelfattah, and Adams [41] affirmed this for Arab regions; and Rafiq, Salim, and Apergis [42] affirmed a positive impact of industrialization on $CO_2$ emissions for 53 countries based on the STIRPAT model. Likewise, Nejat et al. [43] and Tian et al. [44] in China, Salim and Shafiei [45] in 29 OECD countries, and Cherniwchan [46] in 157 countries concluded that industrialization has detrimental effects on $CO_2$ emissions. Besides, Shahbaz et al. [47] found the presence of the EKC hypothesis between industrialization and $CO_2$ emissions in the case of Bangladesh. Asumadu-Sarkodie and Owusu [48] studied the causal link between carbon emissions, energy use, industrialization, and financial development in Sri Lanka from 1971 to 2012. Based on the ARDL approach, the study established a long-run association among the variables. Furthermore, a unidirectional causal link was affirmed between carbon emissions to energy consumption, while a bidirectional causal link was found between industrialization and energy use (a strong determinant of $CO_2$ emissions). Asumadu-Sarkodie and Owusu [48] worked on the causal nexus between $CO_2$ emissions and industrialization for the period 1965–2011 in Rwanda. Based on the ARDL model, the study concluded a long-run association among the variables. Besides, the effect of industrialization on $CO_2$ emissions was found to be positive, adversely affecting health, air quality, and the environment. The study recommends environmentally friendly industrialization policies for Rwanda to reduce $CO_2$ emissions in the country. Zhao et al. [49] used the Log Mean Divisia

Index and concluded that the major factor behind high carbon emissions was industrial output in Shanghai. Appiah et al. [50] identified a causal link between industrialization and $CO_2$ emissions.

Concerning NICs countries, only a few studies (Ghazali & Ali, [51]; Sharif Hossain, [52]; S. Zhang, Liu, & Bae, [53]) have been carried out on the association of carbon emissions with other macroeconomic variables (economic development, urbanization, free trade, energy consumption), given the fact that NICs contribute 42% to global $CO_2$ emissions due to their rapid economic growth (IPCC, 2013) [2]. A summary of the more recent studies on the nexus of financial development, industrialization, and $CO_2$ emissions is provided in Table 2.

**Table 2.** Chronological summary of the literature on industrialization and $CO_2$ emissions.

| Authors | Time Period | Region/Countries | Finding |
| --- | --- | --- | --- |
| Zhao et al. [49] | 1996–2007 | Shanghai | Industrial output positively contributes to $CO_2$ emissions. |
| Zhongping et al. [54] | 1978–2008 | 31 industries in China | Heavy industrialization has a positive effect on $CO_2$ emissions |
| Kivyiro and Arminen [55] | 2000–2012 | Sub-Saharan Africa | Industrialization has a positive impact on $CO_2$ emissions. |
| Al-Mulali and Ozturk [11] | 1962–2012 | 14 MENA countries | A causal link between industrialization, financial development, and $CO_2$ was revealed. |
| Gokmenoglu et al. [1] | 1960–2010 | Turkey | Cointegration exists between financial development, industrialization, and $CO_2$ emissions and unidirectional causality from financial development to $CO_2$ emissions. |
| Xu and Lin [56] | 1990–2011 | 30 provinces in China | Industrialization with $CO_2$ emissions has an inverted U-shaped nonlinear relationship in 3 regions. |
| Sarkodie and Owusu [48] | 1971–2013 | Sri Lanka | A long-run association between industrialization and $CO_2$ emissions was affirmed, while a bidirectional causal link from industrialization to energy consumption was asserted. |
| Sarkodie and Owusu [57] | 1965–2011 | Rwanda | Industrialization accelerates $CO_2$ emissions |
| Appiah et al., [50] | 1990–2014 | Uganda | Industrialization has a positive effect on $CO_2$ emissions. A causal correlation between industrialization and $CO_2$ emissions was also found. |
| Opoku and Aluko [58] | 2000–2016 | 37 African Countries | The industrialization has heterogeneous effects (decrease/increase) for different data quantiles. |
| Rehman et al. [59] | 1971–2019 | Pakistan | Industrialization has positive effects on carbon emissions |

Authors' compilation.

### 2.3. Renewable and Non-Renewable Energy–Pollution Nexus

Numerous studies have examined the impact of renewable and non-renewable energy consumption on environmental quality. For instance, Apergis and Payne [60] analyzed the linkage between REC and $CO_2$ emissions from 1984 to 2007 in the case of 19 developed and developing countries. The Granger causality test findings show that REC does not lead to a short-term reduction in $CO_2$ emissions. In addition, for a group of 12 MENA countries, Farhani [61] explored the relationship between REC and $CO_2$ emissions for the period 1975–2008. The empirical findings show that, in the short-run, one-way causality ranges from REC to $CO_2$ emissions, while the long-term unidirectional causality ranges from $CO_2$ emissions to REC. Bölük and Mert [62] concluded that REC and NREC increase $CO_2$ emissions in the case of 16 European Union (EC) countries. Ben Jebli, Ben Youssef, and Ozturk [63] stated that, in 25 OECD countries, REC decreases $CO_2$ emissions while NREC increases environmental pollution. Bilgili, Koçak, and Bulut [64] found that REC positively reduced carbon emissions for a group of 17 OECD countries over the period 1977–2010. Zoundi [65] and Hu et al. [66] found that REC reduces carbon emissions in the case of 25 sub-Saharan African and 25 developing countries, respectively. Sharif et al. (2019) [67], in a sample of 74 countries, concluded that NREC increases while REC reduces $CO_2$ emissions. Danish, Ulucak, and Khan [68] affirmed the validity of the EKC hypothesis in BRICS countries and found that REC reduces ecological footprints. Similarly, Pham, Huynh, and Nasir [69] found that REC lowers $CO_2$ emissions in 28 Eu-

ropean countries. The overall evidence from past research related to renewable energy and non-renewable energy usage and environmental degradation provides mixed or inconclusive results. This merits further investigation to arrive at a novel conclusion for newly industrialized countries. Table 3 summarizes the recent literature that focuses on the relationship between REC, NREC, and carbon emissions.

**Table 3.** Chronological summary of the literature on REC, NREC, and $CO_2$ emissions.

| Authors | Time Period | Region/Countries | Finding |
|---|---|---|---|
| Shafiei and Salim [45] | 1980–2011 | OECD countries | REC decreases while NREC increases carbon emission. |
| Bilgili, Koçak, and Bulut [64] | 1977–2010 | OECD countries | REC decreases carbon emission |
| Owusu and Asumadu-Sarkodie [48] | 1971–2012 | Sri Lanka | Unidirectional causality found from carbon emissions to NREC |
| Zoundi [65] | 1980–2012 | 25 Sub-Saharan African countries | REC decreases $CO_2$ |
| Bélaïd and Youssef [70] | 1980–2012 | Algeria | NREC has adverse effects, while REC has a positive effect on environmental quality |
| Hu et al. [66] | 1996–2012 | 25 developing countries | Share of REC improves environmental quality |
| Pata [71] | 1974–2014 | Turkey | REC does not affect carbon emissions. |
| Chen et al. [72] | 1980–2014 | China | An increase in NREC increases carbon emissions, while a rise in REC reduces carbon emissions. |
| Danish, Ulucak, and Khan [68] | 1992–2016 | BRICS | REC has a positive impact on environmental quality |
| Pham, Huynh, and Nasir [69] | 1990–2014 | 28 European countries | REC is found to reduce carbon dioxide emissions |
| Salahuddin, et al. [73] | 1984–2016 | 33 Sub-Saharan African countries | Green energy reduces carbon emissions |

Authors' compilation.

In brief, the extant literature shows numerous directions and impact strengths in the relationship between financial development, industrialization, renewable and non-renewable energy consumption, and $CO_2$ emissions. The high variability in results reflects the fact that the studies were performed for different regions, countries, development levels, and variable econometric techniques. In summary, the findings of empirical investigations remain mixed or uncertain, thereby investigating its impact on carbon emissions deserves further deliberation.

## 3. The Theoretical Framework of the Model, Data, and Methodology

### 3.1. Theoretical Framework of the Model

The STIRPAT model propounded by Dietz and Rosa [74] provides this study's theoretical and analytical reference framework. Based on the population, affluence, and technology, the model advocates that population and economic activities are essential sources of $CO_2$ emissions. The general form of the model can be written as follows:

$$I = f(P \ A \ T) \tag{1}$$

The STIRPAT model in nonlinear form can be written as

$$I_{it} = \beta_o P_{it}^{\beta_1} A_{it}^{\beta_2} T_{it}^{\beta_3} \mu_{it} \tag{2}$$

where the subscript ($i = 1, \ldots, N$) indicates the countries, and ($t = 1, \ldots, T$) shows the time span. $\beta_o$ and $\mu_{it}$, reflect country-specific effects and error terms, respectively. While $\beta_1$, $\beta_2$, and $\beta_3$ are the elasticities showing the impact on the environment with respect to P, A, and T (York, 2007) [75].

The STIRPAT model in Equation (2) is converted into a linear form by taking a natural log:

$$lnI_{it} = \beta_0 + \beta_1 lnP_{it} + \beta_2 \ lnA_{it} + \beta_3 lnT_{it} + \mu_{it} \tag{3}$$

Additional to the mentioned variables in Equation (3), in this study, we investigated the impact of economic variables (financial development and trade openness) and green energy (non-renewable and renewable energy) on $CO_2$ emissions. For this, we constructed two models, and these are structures as follows:

$$lnCO2_{it} = \gamma_{ot} + \gamma_{1t}lnY_{it} + \gamma_{2t}lnIO_{it} + \gamma_{3t}\text{lnUR}_{it} + \gamma_{4t}lnFD_{it} + \gamma_{5t}lnTO_{it} + e_{it} \quad (4)$$

$$lnCO2_{it} = \gamma_{ot} \quad + \gamma_{1t}lnY_{it} + \gamma_{2t}lnIO_{it} \\ + \gamma_{3t}\text{lnUR}_{it} + \gamma_{4t}lnFD_{it} + \gamma_{5t}lnTO_{it} + \gamma_{6t}lnNRE_{it} + \gamma_{it}lnRE_{it} + e_{it} \quad (5)$$

In the given models, $CO_2$, Y, IO, and UR represent carbon dioxide emissions, total output, industrial output, and urbanization, representing I, A, T, and P in the STIRPAT model, respectively. The variables $CO_2$, Y, IO, and UR employed for I, A, T, and P are similar to earlier studies Al-Mulali, U. et al. (2015) [11], Saidi and Mbareak (2017) [24], and Appiah et al. (2019) [50]. In terms of additional variables, FD stands for financial development, TO stands for trade openness, NRE stands for non-renewable energy, and RE stands for renewable energy. The impact of these additional variables on $CO_2$ emissions have shown by different studies, where some of them include Shafiei and Salim (2014) [45], Zoundi (2017) [65], Ahmad et al. (2020) [34], Salahuddin et al. (2020) [73], and Khan et al. (2022) [37]. In the specification of the STIRPAT model, carbon emissions are widely used as a dependent variable. Following the studies of Saidi and Mbarek, 2017 [24]; Nasir et al. 2019 [13], and Arshad et al., (2022) [76], we replaced $CO_2$ emissions in metric tons per capita. The $CO_2$ emissions metric tons per capita is used as a dependent variable and the remaining as independent variables. In Equations (4) and (6), the last term $e_{it}$ is the model's error term. Furthermore, the acronym and data source of all variables of the models are described in the Table 4. Data for all variables were retrieved from the World Development Indicators (WDI) for the period 1982–2019. However, the WDI lacked last year's data for renewable and non-renewable energy, which were obtained from the Energy Information Agency (EIA) and British Petroleum, respectively.

**Table 4.** Variables of the study.

| Variables | Description | Source |
|---|---|---|
| Per capita carbon emissions ($CO_2$) | $CO_2$ emissions metric tons per capita | WDI |
| GDP growth (GDPPC) | GDP per capita in USD $ | WDI |
| Urbanization (UR) | Urban population growth in annual % | WDI |
| Financial Development (FD) | Domestic credit to private sector % of GDP | WDI |
| Trade Openness (TO) | the ratio of exports and imports as % of GDP | WDI |
| Industrialization (IO) | Value added by industry (% of GDP) | WDI |
| Renewable energy consumption (RE) | kg of oil equivalent | WDI and US Energy Information Administration (EIA 2020) [77] |
| Non-renewable energy Consumption (NRE) | kg of oil equivalent | WDI and British Petroleum Statistical Review of World Energy (BP 2019) [78] |

### 3.2. Econometric Methodology

Given the variables, including carbon dioxide emission, domestic output, industrial output, urban population, financial development, and trade openness, we tried to establish their cointegration and causal association with the help of the panel cointegration and Granger causality test. Since the various econometrics tests and techniques used in this study are discussed in this section. Firstly, the independence test for cross-section developed by Breusch and Pagan [79] and M. H. Pesaran [80] is briefly discussed. Second, we describe the first- and second-generation panel unit root tests proposed by Levin, Lin, and James Chu [81]; Maddala and Wu [82]; and H. Pesaran [83]. Third, the panel cointegration tests suggested by Pedroni [84] and Kao [85] are introduced. Addition-

ally, the fully modified ordinary least square (FMOLS) method developed by Pedroni [86] is described to estimate the long-run relationship. Finally, the Granger causality tests procedure is introduced to test for the detection of the causal relationship between the study's variables.

### 3.2.1. Cross-Section Dependence Test

Cross-sectional dependence (CD) is one of the most important diagnostics that could be checked before panel cointegration analysis. Normally the problem arises in panel series due to unobserved common factors. The literature proposes that it must be tested before proceeding with the panel unit root to avoid bias and inconsistency in the results [79]. Therefore, first, we attempt to evaluate the CD among model variables by following the Bruech-Pagan LM test (Bruesch-Pagen, [79]) and the Pesaran CD test. Both tests considered the standard model of a panel given in an Equation (6) and tested for interdependence in the model's residuals.

$$y_{it} = \alpha_i + \beta_i x_{it} + \mu_{it} \tag{6}$$

Bruesch-Pagan LM test could be used to test cross-section dependence when the time dimension is greater than the cross-section. This proposes an LM statistic, which is significant for T > N and is given by the following equation:

$$CD_{LM} = T \sum_{i=1}^{N-1} \sum_{j=i+1}^{N} \hat{\rho}_{ij}^2 \tag{7}$$

where CD stands for cross-section dependence, $T$ for time dimension, and $N$ for panel dimension, and $\hat{\rho}_{ij}$ is the sample estimate of the pairwise correlation of the regression residuals ($\hat{\mu}_{ij}$), and associated with Equation (6). This $\hat{\rho}_{ij}$ is calculated with the formula given below:

$$\hat{\rho}_{ij} = \frac{\sum_{t=1}^{T} \hat{\mu}_{it} \, \hat{\mu}_{ji}}{\left(\sum_{t=1}^{T} \hat{\mu}_{it}^2\right)^{1/2} \left(\sum_{t=1}^{T} \hat{\mu}_{ji}^2\right)^{1/2}} \tag{8}$$

The stated CDLM test is asymptotically distributed as Chi-square with N(N − 1)/2 degree of freedom under the null hypothesis, i.e., no relation exists between the cross-sections.

Pesaran [80] improved this test and considered when the time dimension is smaller than the cross-section or panel dimension. Pesaran proposed alternative CD test statistics as follows:

$$CD_P = \sqrt{\frac{2T}{N(N-1)}} \sum_{i=1}^{N-1} \sum_{j=i+1}^{N} \hat{\rho}_{ij} \tag{9}$$

This test is applicable even N > T, and the null hypothesis is similar to that of the CDLM test.

### 3.2.2. Panel Unit Root Test

Next to the cross-section dependence test, the panel unit root must be tested to proceed with the cointegration analysis. The literature has divided panel unit root tests into two "first-generation" and "second-generation" tests. The first-generation tests could be used if we find no evidence for cross-section dependence, while the second-generation tests could be employed if we find evidence for cross-section dependence. The LLC and WU tests are the most widely applicable tests of the first-generation tests. The LLC test was developed by Levin, Lin, and Chu [81], while Maddala and Wu [82] introduced MW. Both tests are the extensions of the augmented Dickey–Fuller (ADF) test, under the restrictive assumption of "no relation between individual cross-section". For a panel data set, the LLC test is used as a modification of the ADF regression as follows:

$$\Delta Y_{i,t} = \rho_i Y_{i,t-1} + \sum_{k=1}^{n} \varnothing_i k \Delta Y_{i,t-k} + X'_{it}\delta + \mu_{it} \tag{10}$$

In which, $Y_{it}$ ($i$= 1, 2, ..., $n$; $t$ = 1, 2, ..., T) is a series for a country i over period t. The residual μit is white noise~N(0; σ2) and is assumed to be cross-section independent. The LLC test is conducted under the null hypothesis that "panels contain unit roots", while the alternative hypothesis indicates that "panels are stationary". The hypothesis may be written as follows:

$$H_0 : \rho_i = \rho = 0$$

$$H_1 : \rho_i = \rho < 0$$

Maddala and WU (MW, 1999) [82] is another panel unit root test of the "first generation". It is a non-parametric test and proposes a Fisher-type test that combines the logs of the *p*-values of each cross-section unit root test. The test statistics of MW have a $\chi^2$ distribution with 2n degrees of freedom, where *n* represents the total countries in the panel. The following equation gives the MW test statistic:

$$\lambda = -2 \sum_{i=1}^{n} log_e(\rho_i) \sim \chi^2 2nd(d.f) \tag{11}$$

where $p_i$ stands for *p*-values from the test of ADF unit root for each unit *i*. Maddala and Wu (1999) [82] found that the test has certain advantages over the IPS panel unit test. It is superior to IPS because it does not depend on various lag lengths in the individual or cross-section ADF regressions.

In addition, the unit root is tested using a cross-sectional ADF (CADF). This is a notable test of the "second generation" of unit roots, which was proposed by Pesaran [83]. This test is popular to allow for the existence of cross-sectional dependence. Further, the test could be applied with N > T and gives strong results when T > N, and the test is based on the mean of individual ADF-test statistics of each unit in the panel. CADF test applies unit root for every cross-section forming panel and for the panel itself. The extended ADF regression of the test can be written as follows:

$$\Delta y_{it} = a_i + \rho_i y_{i,t-1} + \sum_{k=1}^{p} b_{ik}\Delta y_{i,t-k} + c_i \bar{y}_{t-1} + \sum_{k=0}^{p} d_{i,k}\Delta \bar{y}_{t-k} + \varepsilon_{i,t} \tag{12}$$

The null hypothesis assumes that all series contain a unit root, symbolically as, $H_0 : \rho_i = 0$ for all i . However, in the alternative hypothesis assume that at least one of the cross-sections in the panel is stationary, as symbolically, $H_1 : \rho_i < 0$ for at least one i.

### 3.2.3. Panel Cointegration Test

Two alternative approaches to panel cointegration have been used to provide more reliable estimates in cointegration testing. Panel cointegration tests show whether there is a linear association between the dependent and independent variables of the model in our panel or not. We employ two important panel cointegration tests, including the Pedroni (1999) [84] and Kao (1999) [85] tests. Both are residual-based cointegration tests that extend the Engle–Granger (EG) framework to test panel data. The Pedroni cointegration approach proposes several tests, and one can calculate the Pedroni's within the dimension and between dimension ADF and PP test statistics. We would also estimate these two in our study, as Pedroni [84] suggests that these two test statistics have the best properties if the samples have a small time dimension. The test's estimation method is an extension of the traditional EG methodology. Therefore, in two steps, estimate the desired panel cointegration. In the starting point, the dependent variable is regressed on explanatory variables, such as in our case, the cointegration equation specified as:

$$lny_{it} = a_i + \beta X_{i,t} + \epsilon_{i,t} \tag{13}$$

In this panel regression, for $t$ = 1, 2, … T, $i$ = 1, 2, … , N; where αi is constant or individual effects, y is the dependent variable ($CO_2$), and X is a vector of explanatory regressors includes domestic income (Y), industrial output (IO), urban population (UR), financial development (FD), trade openness (TO), non-renewable energy (NRE), and renewable energy (RE). The last term $\epsilon_{i,t}$ is a white noise term. Since then, after estimating Equation (8), the stationarity of $\widehat{\epsilon}_{i,t}$ is examined with either the Dickey–Fuller test or the Phillips–Perron test. For the hypothesis test, the structure of estimated residuals may be as given, $\epsilon_{i,t} = \rho\epsilon_{i,t-1} + \varepsilon_{it}$. Pedroni describes the null hypothesis of no cointegration ($\rho_i = 1$); however, the author proposes two alternative hypotheses, i.e., the homogenous alternative as $\rho_i = \rho < 1$ for all i (for panel statistic test or within dimension), and the heterogeneous alternative as $\rho_i < 1$ for all i (for the group statistic or between dimensions). Finally, the decision of cointegration is made if the test statistic value rejects the null hypothesis and accepts the alternative hypothesis.

Kao [85] also tests the Kao test of panel cointegration. Kao proposes different DF-type and ADF-type tests to test the null hypothesis of no cointegration. This approach tests for cointegration in homogenous panels. In Kao tests, it is assumed that all the cointegration vectors in every cross-section are identical.

### 3.2.4. Fully Modified OLS

As a next step, we used a fully modified ordinary least square to determine the long-run relationship between variables of the model in Equation (3). Phillips and Hansen [87] first developed FMOLS to provide an optimal estimate of the cointegrating regression model. However, we estimate long-run relationships using the FMOLS suggested by Pedroni [86]. According to Ozcan [88], FMOLS has the advantage of correcting both the serial correlation and endogeneity bias. The FMOLS is the most appropriate estimation technique if the panel contains heterogeneous cointegration. Most importantly, using this approach, more consistent and unbiased estimates are generated even in small samples where the time dimension is not less than the cross-section (Pedroni, 2001) [86]. In this respect, FMOLS assume a regression of the panel as given:

$$y_{i,t} = a_i + \beta x_{i,t} + \mu_{i,t} \tag{14}$$

$$x_{i,t} = x_{i,t-1} + \tilde{\xi}_{i,t} \tag{15}$$

In which $a_i$ is country-specific fixed effects, and β is a cointegrating vector. Therefore, the $\varepsilon_{i,t} = (\mu_{i,t} + \tilde{\xi}_{i,t})$ vector error process is also a stationary process. Using relevant notations, the FMOLS estimator for group means can be defined as:

$$\hat{\beta}*_{GFM} = N^{-1}\sum_{i=1}^{N}\left(\sum_{t=1}^{T}(x_{i,t} - \overline{x}_{i,})^2\right)^{-1}\left(\sum_{t=1}^{T}(x_{i,t} - \overline{x}_i)y*_{i,t} - T\hat{\gamma}_i\right) \tag{16}$$

where, $y*_{i,t} = (y_{i,t} - \overline{y}_i) - \frac{\hat{\Omega}_{21i}}{\hat{\Omega}_{22i}}\Delta y_{i,t}$; $y*_{i,t}$ is the transform variable of $y_{it}$; $\hat{\gamma}_i = \hat{\Gamma}_{21i} + \hat{\Omega}^o_{21i} - \frac{\hat{\Omega}_{21i}}{\hat{\Omega}_{22i}}(\hat{\Gamma}_{22i} + \hat{\Omega}^o_{22i})$; and $\hat{\gamma}_i$ is the serial correlation correction term.

### 3.2.5. Panel Granger Causality Analysis

It is pertinent to mention that the purpose of the cointegration relationship is not to indicate the direction of the causal relationship among variables of the model. The causality tests could be used to identify the direct causal linkages, which are normally carried out after the appearance of cointegration. In this research, we analyze causality through the pairwise directions by the Granger causality tests for panel data introduced by Engle and Granger [89]. Additionally, we employ the DH causality approach proposed by Dumitrescu and Hurlin [90]. This DH approach offers more useful information and is

suitable for analyzing the panel's cross-sectional dependencies. The Granger equation for the panel data model is considered as follows:

$$\Delta Y_{i,t} = a_i + \sum_{i=1}^{k} \varphi_i^{(k)} Y_{i,t-k} + \sum_{i=1}^{k} \beta_i^{(k)} X_{i,t-k} + \varepsilon_{it} \tag{17}$$

where, in Equation (15), k denotes optimum lags, $Y$ and $X$ are the variables in which causality will be estimated, the intercept ($\alpha$) represents fixed individual effects, $\varphi_i^{(k)}$ and $\beta_i^{(k)}$ respectively denote autoregressive parameters and regression coefficients, and the last term $\varepsilon$ is the white noise error term. This test was used to detect whether or not pairwise causality exists between X and Y. As stated above, it should be noted that the Granger test coefficients are assumed to be the same across countries, while DH assumes differences across the cross-sections [88,89]. Both approaches have the same null hypothesis, i.e., H$_0$: $X$ does not cause $Y$.

## 4. Empirical Analysis

The empirical analyses were carried out for the balanced panel dataset, and all variables were transformed into a natural logarithm. The results are reported in the following sub-sections accordingly.

### 4.1. Descriptive Statistics, Correlation Matrix, and Cross-Section Dependency Test

The study used descriptive statistics, a correlation matrix, and a cross-section dependence test before applying the panel unit root. The descriptive statistics and correlation matrix were conducted to describe the characteristics of the model's variables and correlation analysis, and their results are shown in Table 5.

**Table 5.** Descriptive statistics and correlation matrix results of study variables' descriptive statistics.

| Variables | Mean | Std. Dev. | | Skew | Kurt | J-B | | Obs. |
|---|---|---|---|---|---|---|---|---|
| lnCO$_2$ | 3.554 | 2.660 | | 0.928 | 2.769 | 51.207 * | | 351 |
| lnY | 7.788 | 1.054 | | −0.482 | 2.424 | 18.472 * | | 351 |
| lnIO | 3.495 | 0.212 | | −0.172 | 2.378 | 7.391 ** | | 351 |
| lnUR | 3.903 | 0.383 | | −0.492 | 2.119 | 25.528 * | | 351 |
| lnFD | 3.937 | 0.748 | | −0.160 | 1.772 | 23.561 * | | 351 |
| lnTO | 3.902 | 0.674 | | 0.119 | 2.479 | 4.797 *** | | 351 |
| lnNRE | 4.289 | 0.242 | | −0.844 | 2.641 | 43.564 * | | 351 |
| lnRE | 3.095 | 0.692 | | −0.640 | 3.178 | 24.431 * | | 351 |

**Correlation Matrix**

| Variables | lnCO$_2$ | lnY | lnIO | lnUR | lnFD | lTO | lRE | lNRE |
|---|---|---|---|---|---|---|---|---|
| lnCO$_2$ | 1 | | | | | | | |
| lnY | 0.582 | 1.000 | | | | | | |
| lnIO | 0.413 | 0.844 | 1.000 | | | | | |
| lnUR | 0.527 | 0.236 | −0.072 | 1.000 | | | | |
| lnFD | 0.361 | 0.376 | 0.128 | 0.439 | 1.000 | | | |
| lnTO | 0.186 | −0.160 | −0.253 | 0.457 | 0.432 | 1.000 | | |
| lnRE | −0.696 | −0.668 | −0.490 | −0.315 | −0.651 | −0.324 | 1 | |
| lnNRE | 0.719 | 0.553 | 0.312 | 0.345 | 0.476 | 0.271 | −0.826 | 1 |

Note: Table 5 reports mean, standard deviation (Std. Dev.), skewness (Skew), kurtosis (Kurt), and Jarque–Bera (J-B) test statistics for respective variables. The *, **, and *** represent 1%, 5%, and 10% levels of significance, respectively. In the first column, ln stands a natural log, and in the last column, obs. stands for observation.

In Table 5, it is obvious that the mean value of all variables, except the income level and non-renewable energy, is less than 4. The $CO_2$ variable, however, shows a wide dispersion as its standard deviation is relatively greater and has a value of 2.66. The skewness value is negative for all variables except $CO_2$ and trade openness (TO). The negative values show a skew towards the left while the positive skew towards the right. The positive and greater skewness value for $CO_2$ than zero indicates that the right tail is longer than the left. Kurtosis is positive for all variables, indicating a fat-tailed phenomenon.

Additionally, the results of the J-B tests look significant, showing that the data for all variables are not from the normal distribution. Furthermore, results show that all variables positively correlate with carbon emissions except renewable energy. Renewable energy is negatively correlated with carbon emissions. Notably, income level, renewable energy, and non-renewable energy show a high correlation with carbon emissions, whereas industrial output shows relatively less correlation.

Proceeding further, the model's cross-sectional dependency is tested. Two tests were used for this purpose, i.e., Breusch and Pagan (1980)'s [79] LM and M. H. Pesaran (2004)'s [80] CD tests. The models' results are based on a fixed effect estimator, derived from Equations (4) and (5), in which $CO_2$ is a dependent variable and Y, IO, UR, FD TO, NRE, and RE are explanatory variables. The test results are shown in Table 6.

**Table 6.** Cross-sectional dependency test results.

| Test | Model-1 (Excludes Green Energy) | | Model 2 (Includes Green Energy) | |
| --- | --- | --- | --- | --- |
| | Test-Statistic | *p*-Value | Test-Statistic | *p*-Value |
| **Bruesch–Pagen LM** | 245.82 | 0.000 | 388.76 | 0.000 |
| **Pesaran CD** | −3.46 | 0.000 | −3.42 | 0.000 |

Note: $H_O$: no cross-section dependence. Significance level; $p < 0.001$.

As we can see, the *p*-value is significant in the case of both LM and CD tests, indicating a strong cross-sectional dependence. Pesaran's test, however, could normally be used when the cross-section is larger than time. In our case, therefore, the LM test should be taken as T is > than N. This suggests that it is safe to reject the null and accept the alternative hypothesis and concludes that our models contain cross-sectional dependence under a fixed-effects assumption. We can also conclude that second-generation panel unit roots would produce a more consistent result than first-generation panel unit roots.

*4.2. Panel Unit Root Results*

Table 5 reports the results of panel unit root tests at the level and first difference for all study variables. In this paper, we applied unit root tests from both the "first-generation" and "second-generation" viewpoints. As discussed in the Methodology Section, we tested unit roots using LLC, MU, and CADF. Table 5 shows that the results of the panel unit tests are mixed, especially in the case of LLC. In the case of first-generation unit roots tests, the LLC test indicates that some of the variables are stationary at both levels and the first difference.

Similarly, the MU test suggests that all panels are non-stationary at the level, except the urbanization and non-renewable energy. However, all panels are stationary at the first difference, even at a 1% significance level. In the second-generation test case, the $CO_2$, income level (Y), and urbanization (UR) are stationary at a 5% significance level, while the other variables are non-stationary. In the first difference, except for the urban population (UR), CADF suggests stationarity with a strong significance level at the first difference. Consequently, all variables are first difference stationary. However, because of the clear cross-sectional dependency reported in Table 6, we decided to give more importance to the CADF test relative to the IPS and MU. Table 7 represents the panel unit root test statistics.

### 4.3. Results of the Panel Cointegration Test

This was found in unit root results, that all panel series are stationary at first difference. The panel cointegration is therefore tested to determine whether or not cointegration exists between the model variables. First, we employed Pedroni's panel cointegration [86]. The model includes intercept only and intercept and trend, and no-cointegration is assumed as the null hypothesis. The test results are reported in Table 8. The estimated results of Pedroni's test suggest rejecting the null hypothesis and accepting the alternative, supporting the presence of cointegration in cases of "intercept only" and "intercept and trend". In the first case, only two tests, i.e., ADF and PP, confirm cointegration with a significance level of 1%. However, in the second case, even modified PP test results conclude that the cointegration relation between variables is significant at 5%. We have also tested Kao panel cointegration, where Kao introduced parametric residual-based panel cointegration. Thus, Kao's test results also show that model variables are cointegrated and have a long-run relationship, irrespective of the test statistics (see Table 9). There is enough evidence to reject the null hypothesis of no cointegration at a 1% significance level since the results show that cointegration exists between variables.

**Table 7.** Results of panel unit root test statistics.

| Variables | LLC | | CADF Unit Root | | MW Unit Root | |
|---|---|---|---|---|---|---|
| | Level | 1st-Diff. | Level | 1st-Diff. | Level | 1st-Diff. |
| $lnCO_2$ | 0.77 (0.781) | −7.55 * (0.000) | −2.10 ** (0.018) | −12.11 * (0.000) | 11.11 (0.889) | 320.99 * (0.000) |
| lnY | 1.23 (0.891) | −8.08 * (0.000) | −1.79 ** (0.037) | −11.42 (0.000) | 2.21 (1.000) | 187.06 * (0.000) |
| lnIO | −2.05 (0.019) ** | −7.19 * (0.000) | 0.59 (0.725) | −11.13 * (0.000) | 24.580 (0.136) | 250.66 * (0.000) |
| lnUR | 0.65 (−0.743) | −2.99 * (0.001) | −4.52 * (0.000) | 0.89 (0.812) | 337.02 * (0.000) | 16.72 (0.542) |
| lnFD | −1.37 (0.085) *** | −7.62 * (0.000) | 0.15 (0.561) | −9.97 * (0.000) | 18.49 (0.417) | 216.92 * (0.000) |
| lnTO | −1.66 (0.048) ** | −8.61 * (0.000) | −1.99 (0.139) | −8.54 * (0.000) | 19.81 (0.343) | 203.38 * (0.000) |
| lnNRE | −3.66 (0.000) | −7.61 (0.000) | −1.58 (0.057) | −11.11 * (0.000) | 29.71 (0.040) ** | 297.61 * (0.000) |
| lnRE | 2.64 (0.996) | −7.46 (0.000) | −1.47 (0.071) | −10.09 * (0.000) | 4.38 (0.999) | 255.13 * (0.000) |

Note: A model with a constant for all variables was selected as a test model. The bracket values are *p*-values. The *, ** and *** indicate significance levels at 1% 5% and 10%, respectively. The MU test results are based on Chi-square test statistics. In the CADF test suggested by Pesaran, $H_0$ for all tests is that the variables are I(1), and used the STATA routine pescadf. In Maddala and Wu (MW) [82], $H_0$ is the unit root and we used the STATA routine xtfisher.

**Table 8.** Pedroni test results for cointegration (Pedroni, 1999 and 2001) [84,86].

| Test Statistic | Included Intercept | Included Intercept and Trend |
|---|---|---|
| **Modified Phillips-Perron t** | −1.173 (0.120) | 2.168 ** (0.015) |
| **ADF t Statistics** | −3.371 * (0.000) | −4.379 * (0.000) |
| **PP t Statistics** | −3.309 * (0.000) | −3.774 * (0.000) |

Note: the null hypothesis is that the variables are not cointegrated, while the alternative assumes 'that all panels are cointegrated. The * and ** indicate the rejection of the null hypothesis at 1% and 5%, respectively.

**Table 9.** Kao test results for panel cointegration (Kao, 1999). [85].

| Test Statistic | Statistic | *p*-Value |
|---|---|---|
| **Modified Dickey-Fuller** | −2.366 * | 0.009 |
| **Dickey-Fuller t** | 1.740 ** | 0.041 |
| **Augmented Dickey-Fuller t** | −1.953 ** | 0.025 |
| **Unadjusted modified Dickey-Fuller t** | −2.200 ** | 0.014 |
| **Unadjusted Dickey-Fuller t** | −1.672 ** | 0.047 |

Note: The model includes constant only. The null hypothesis is that the variables are not cointegrated, while the alternative. The * and ** indicate the rejection of the null hypothesis at 1% and 5%, respectively.

### 4.4. Fully Modified OLS Results

Given that the selected model variables are cointegrated, next to that we estimated the long-run relationship. For this, we employed Pedroni's FMOLS estimator and included the only constant in the cointegration relationship using FMOLS. The study variables are stationary in a different order of integration. Therefore, FMOLS is used as a robust test. The FMOLS test results are reported for both pooled and weighted estimation in Table 10.

**Table 10.** Panel of fully modified OLS results.

| | Dependent Variable = LN_CO$_2$ | | | |
|---|---|---|---|---|
| **Method** | **FMOLS (Pooled)** | | **FMOLS (Weighted)** | |
| **Variable** | **Coefficient** | **Coefficient** | **Coefficient** | **Coefficient** |
| **lnY** | 0.603 *( 3.25) | 0.061 (0.34) | 0.519 * (35.47) | 0.076 * (3.47) |
| **lnIO** | 1.901 * (4.16) | 1.257 * (3.14) | 1.728 * (77.75) | 1.408 * (57.31) |
| **lnUR** | 5.746 * (7.74) | 4.912 * (7.13) | 5.747 * (1744.70) | 4.739 * (729.02) |
| **lnFD** | −0.292 (−1.42) | −0.368 ** (−2.06) | −0.197 * (−25.16) | −0.307 * (−30.91) |
| **lnTO** | −0.884 (−3.67) | −0.845 * (−3.63) | −0.634 * (−29.94) | −0.778 * (−83.18) |
| **lnNRE** | | 0.885 (1.38) | | 0.687 * (41.09) |
| **ln RE** | | −1.599 * (−5.58) | | −1.560 * (−111.72) |
| **R$^2$** | 0.958 | 0.968 | 0.958 | 0.967 |
| **Adj. R$^2$** | 0.956 | 0.966 | 0.957 | 0.966 |

Note: the * and ** indicate 1% and 5% significance levels, respectively. The bracket values are *t*-test values. Only intercept is included in the model. Bartlett kernel with Newey–West fixed bandwidth is used.

To estimate the specified regression given in Equations (4) and (5), we ran FMOLS, and the results are reported in Table 10. Almost all coefficient estimates are statistically significant, regardless of the panel method. It should be noted that Pedroni (2001) [86] proposed pooled weighted FMOLS estimates. When the models are estimated with a weighted panel method, the elasticity coefficient varies from −01.59 to 4.91 and from −1.56 to 4.73, respectively. Irrespective of the models, the main results confirm the significant positive relationship between domestic income level and CO$_2$, industrial output and CO$_2$, and urban population and CO$_2$. The results show that the Y, IO, and UR coefficient values are in the ranges 0.06–0.60, 1.25–1.90, and 4.73–5.75, respectively. These positive results are consistent with the findings of various previous studies, see for instance (Al-mulali et al. [11]; Salahuddin & Gow, [91]; Yang et al. [16]). Instead, Yao et al. [92] and Shuai et al. [93] found a negative association between urbanization and emissions of CO$_2$. In contrast, there is a negative relationship between financial development and CO$_2$ and trade openness and CO$_2$, and the coefficient values in pooled estimation are −0.36 and −0.84 for financial development and trade openness, respectively. These negative effects of the FD are consistent with studies conducted by Khan et al. [37], and our results corroborated with Sinha and Shahbaz [94], Afridi et al. [95], Syed [96], and Syed et al. [97] for trade openness. Furthermore, the results from FMOLS (pooled) indicated that NRE is positive but insignificant in explaining the CO$_2$ of the panel. If we consider this in FMOLS (weighted), then non-renewable energy is positive and significant in explaining the CO$_2$ of the panel. This implies that non-renewable energy is insufficient in reducing CO$_2$ in selected nine new industrialized countries. Instead, the coefficient of renewable energy (RE) is negative and significant in both cases. The coefficient of RE is around 1.59 and significant at the 1% level, indicating that a 1% increase in RE contributes to reducing CO$_2$ emissions by 1.59% in the long run for nine NICs.

Proceeding further, it is observed that a strong positive relationship runs from GDP per capita, industrial output, and urbanization to CO$_2$ emissions, and a negative relationship runs from financial development and trade openness to CO$_2$ emissions. The coefficient estimates are highly elastic for urbanization in the models. It is also elastic for industrial output and renewable energy, whereas it is less elastic for financial development. Hence,

by applying the method of FMOLS, we investigated the long-run relationship between variables of the model. Our methodology is in line with the study by Neagu [98] on European countries that used FMOLS (see also, Mitić et al. [99]). However, for the robustness and validity of the results, we extracted the residuals and tested them for white noise to ensure that our long-run model results are not spurious (see Table 11). The results indicate that in both cases, the residuals are stationary at that level, indicating that our results are consistent and unbiased and that the long-run regression model is not spurious.

**Table 11.** Test for residuals of regression model.

| Model | Method | Statistic | Prob. |
|---|---|---|---|
| FMOLS (Pooled) | LLC test | −4.212 | 0.000 |
| FMOLS (Weighted) | LLC test | −4.304 | 0.000 |

### 4.5. Panel Causality Tests Results

The directions of causality are tested using the Engle–Granger approach and DH approach. Table 12 represents the results of Equation (5) causality tests, including all the variables that have causality with $CO_2$ emissions. The full test pairwise significant results are given in Appendix A. If the *p*-values of the respective test statistic are significant at 1% and 5%, then reject the null hypothesis of non-causality and accept the alternative. In Granger causality, industrial output, urbanization, financial development, trade openness, non-renewable energy, and renewable energy are significant at 1%, while GDP per capita is significant at 10%, so we reject $H_0$, which indicates unidirectional causality.

**Table 12.** Panel causality test results (Granger causality and Dumitrescu–Hurlin).

| Statement of Null Hypothesis | Granger Causality | | Dumitrescu–Hurlin | | |
|---|---|---|---|---|---|
| | F-Stat | Prob. | W-Stat | Zbar Stat | Prob. |
| Y does not Granger cause $CO_2$ | 2.628 *** | 0.073 | 4.634 * | 3.288 * | 0.001 |
| $CO_2$ does not Granger cause Y | 0.097 | 0.907 | 1.893 | 0.315 | 0.752 |
| IO does not Granger cause $CO_2$ | 5.143 * | 0.006 | 5.801 * | 4.824 * | 0.000 |
| $CO_2$ does not Granger cause IO | 0.959 | 0.384 | 2.697 | 9.741 | 0.458 |
| UR does not grange cause $CO_2$ | 4.679 * | 0.009 | 4.494 * | 3.106 * | 0.002 |
| $CO_2$ does not Granger cause UR | 1.329 | 0.266 | 5.462 * | 4.379 * | 0.000 |
| FD does not Granger cause $CO_2$ | 6.930 * | 0.008 | 2.900 | 1.009 | 0.313 |
| $CO_2$ does not Granger cause FD | 0.775 | 0.379 | 3.275 | 1.503 | 0.133 |
| TO does not Granger cause $CO_2$ | 5.642 * | 0.004 | 4.421 | 2.974 * | 0.003 |
| $CO_2$ does not Granger cause TO | 0.907 | 0.405 | 3.254 *** | 1.578 *** | 0.094 |
| NRE does not Granger cause $CO_2$ | 4.749 * | 0.009 | 4.409 * | 2.994 * | 0.0003 |
| $CO_2$ does not Granger cause NRE | 0.738 | 0.478 | 2.602 | 0.617 | 0.537 |
| RE does not Granger cause $CO_2$ | 4.988 * | 0.007 | 4.426 * | 3.016 * | 0.002 |
| $CO_2$ does not Granger cause RE | 0.607 | 0.512 | 3.301 | 1.536 | 0.124 |

Note: the * and *** denote that the null is rejected at 1%, and 10%, respectively.

Similarly, the DH causality test results show that GDP per capita (Y), industrial output (IO), non-renewable energy (NRE), and renewable energy (RE) are significant at 1%, indicating unidirectional causality from GDP per capita, industrial output, non-renewable energy, and renewable energy to $CO_2$ emissions. Urbanization (UR) has significant values of 1%, so we reject the null hypothesis, and the results indicate a bidirectional causality relationship between urbanization and $CO_2$ emissions. The $CO_2$ and trade openness (TO) are significant at 10%, indicating weak bidirectional causality. The DH test results are insignificant for financial development (FD) and $CO_2$, so accept the null hypothesis of non-causality. Consequently, unidirectional causality was found between domestic output, industrial output, non-renewable energy, and $CO_2$ emissions, while bidirectional causality

runs from urbanization and renewable energy to $CO_2$ emissions. The causality between $CO_2$ emissions and financial development is not significant. For complete significant results of the panel causality tests see Appendix A.

## 5. Conclusions and Policy Implications

The prime objective of the study is to find the long-run estimates of $CO_2$ emissions and the causal relationship between industrial output, financial development, gross domestic product, trade openness, non-renewable energy, renewable energy, and $CO_2$ emissions for the panel of NIC countries using data for the period from 1980 to 2019. To the best of our knowledge, only a few studies have investigated the causal relationship between carbon emissions and variables like GDP, energy consumption, trade openness, and urbanization in the case of new industrialized countries. In this research, the omitted variable bias is addressed by focusing on variables like industrial production and financial development, which have attained little attention in previous research work. Furthermore, previous research has concentrated on the energy use effects without considering the energy consumption structure (i.e., renewable and conventional energy). Thus, our study covers both forms of energy usage, which can provide essential information for successful energy policies that contribute to attaining long-term development goals.

We tested the model's variables for cross-sectional dependence, unit root, and long-run relationship before estimating for the long-term effects of GDP, energy consumption, trade openness, and urbanization on carbon emissions. The findings show that all variables become stationary after the first difference and that the model variables in the panel of newly industrialized countries have a long-run relationship. Economic growth, industrial output, urbanization, and non-renewable energy have long-term positive effects on $CO_2$ emissions, while financial development, trade openness, and renewable energy have negative effects on $CO_2$ emissions during the period of analysis. Furthermore, industrial output, urbanization, and renewable energy strongly influence $CO_2$ emissions in the countries analyzed.

Additionally, a significant bidirectional causality was found between $CO_2$ emissions and urbanization. Instead, significant pairwise unidirectional causality was found between domestic output and $CO_2$, industrial output and $CO_2$, non-renewable energy and $CO_2$, and renewable energy and $CO_2$ emissions, while insignificant unidirectional causality between $CO_2$ emissions and financial development. The results indicate that increasing effects of industrial output and urbanization rises $CO_2$ emissions, while the increase in trade openness and renewable energy leads to reducing $CO_2$ emissions.

The study results propose some policy implications as follows. The industrial output has a positive relationship with carbon emissions. Therefore, governments of newly industrialized nations can set emission standards to control carbon dioxide and other pollutants during production. Furthermore, it is suggested that governmental and commercial sectors focus on ecologically friendly, contemporary processing, and efficient production methods. Likewise, the newly industrialized countries must use environmentally friendly resources as their inputs to achieve economic growth without degrading the environment. Financial development has a negative association with emissions of $CO_2$. The study recommended that newly industrialized countries extend their financial resources to achieve high economic output.

Future research on the theme should focus on the variables, such as democracy and population, as well as the relationship between industrial production, financial development, and the environment, using disaggregated data from other developing/emerging blocs. Finally, circumstances in different developing countries may differ significantly. As a result, future research should examine broadening the analysis to include the cases of other emerging areas, such as South Asia, East Asia, and the Pacific; the Middle East and North Africa; and Latin America and the Caribbean. Such extensive evaluations can significantly contribute to reaching a broad conclusion about the influence of industrial activity and financial development on $CO_2$ emissions.

**Author Contributions:** Conceptualization, S.P., S.K., M.A.K., M.A.A. and A.A.S.; methodology, S.P., S.K., M.A.K., M.A.A., A.A.S. and S.G.; software, S.G.; validation, S.P., S.K., M.A.K., M.A.A., A.A.S. and S.G.; formal analysis, S.P., S.K., M.A.K., M.A.A., A.A.S. and S.G.; investigation, S.P., S.K., M.A.K., M.A.A., A.A.S. and S.G.; resources, S.P., S.K., M.A.K., M.A.A. and A.A.S.; data curation, S.P., S.K., M.A.K., M.A.A. and A.A.S.; writing—original draft preparation, S.P., S.K., M.A.K., M.A.A. and A.A.S.; writing—review and editing, S.G.; visualization, S.P., S.K., M.A.K., M.A.A., A.A.S. and S.G.; supervision, S.G.; project administration, A.A.S. and S.G. All authors have read and agreed to the published version of the manuscript.

**Funding:** This research received no external funding.

**Institutional Review Board Statement:** Not applicable.

**Data Availability Statement:** The data supporting the study's findings are available from the corresponding author upon reasonable request.

**Conflicts of Interest:** All authors declare no conflict of interest in this paper.

## Appendix A  Complete Significant Results of the Panel Causality Tests

**Table A1.** Panel Granger causality and DH causality test results for all variables.

| Null Hypothesis Test | Granger Causality Test | | Dumitrescu–Hurlin Causality Tests | | |
| --- | --- | --- | --- | --- | --- |
| | F-Stat | *p*-Value | W-Stat. | Zbar-Stat. | *p*-Value |
| Y does not Granger cause $CO_2$ | 2.628 *** | 0.074 | 4.633 * | 3.288 * | 0.001 |
| IO does not Granger cause $CO_2$ | 5.1423 * | 0.006 | 5.801 * | 4.824 * | 0.000 |
| UR does not Granger cause $CO_2$ | 4.679 * | 0.009 | 4.494 * | 3.106 * | 0.002 |
| $CO_2$ does not Granger cause UR | 1.329 | 0.266 | 5.463 * | 4.379 * | 0.000 |
| TO does not Granger cause $CO_2$ | 5.642 * | 0.004 | 4.421 * | 2.974 * | 0.003 |
| NRE does not Granger cause $CO_2$ | 4.749 * | 0.009 | 4.409 * | 2.994 * | 0.003 |
| RE does not Granger cause $CO_2$ | 4.987 * | 0.007 | 4.426 * | 3.016 * | 0.002 |
| Y does not Granger cause IO | 4.802 * | 0.008 | 3.739 ** | 2.112 ** | 0.035 |
| UR does not Granger cause Y | 1.984 | 0.139 | 9.500 * | 9.691 * | 0.000 |
| Y does not Granger cause UR | 3.681 ** | 0.026 | 5.606 * | 4.568 * | 0.000 |
| Y does not Granger cause FD | 3.789 ** | 0.024 | 5.568 * | 4.518 * | 0.000 |
| TO does not Granger cause Y | 2.409 | 0.091 | 8.016 * | 7.739 * | 0.000 |
| Y does not Granger cause TO | 0.331 | 0.718 | 5.691 * | 4.679 * | 0.000 |
| NRE does not Granger cause Y | 1.022 | 0.3609 | 3.678 ** | 2.033 ** | 0.042 |
| RE does not Granger cause Y | 1.241 | 0.290 | 3.859 ** | 2.269 ** | 0.023 |
| Y does not Granger cause RE | 0.435 | 0.647 | 3.706 ** | 2.069 ** | 0.039 |
| UR does not Granger cause IO | 3.554 ** | 0.029 | 5.639 * | 4.612 | 0.000 |
| FD does not Granger cause IO | 9.340 * | 0.000 | 5.788 * | 4.808 * | 0.000 |
| IO does not Granger cause FD | 0.179 | 0.836 | 4.143 * | 2.644 * | 0.008 |
| IO does not Granger cause TO | 0.312 | 0.732 | 4.169 * | 2.678 * | 0.007 |
| IO does not Granger cause NRE | 0.172 | 0.842 | 5.622 * | 4.589 * | 0.000 |
| IO does not Granger cause RE | 1.217 | 0.297 | 4.288 * | 2.834 * | 0.005 |
| FD does not Granger cause UR | 4.055 ** | 0.018 | 5.229 * | 4.073 * | 0.000 |

Table A1. *Cont.*

| Null Hypothesis Test | Granger Causality Test | | Dumitrescu–Hurlin Causality Tests | | |
|---|---|---|---|---|---|
| | F-Stat | *p*-Value | W-Stat. | Zbar-Stat. | *p*-Value |
| UR does not Granger cause FD | 0.206 | 0.814 | 4.284 * | 2.829 * | 0.005 |
| TO does not Granger cause UR | 1.518 | 0.221 | 4.768 * | 3.466 * | 0.000 |
| UR does not Granger cause TO | 0.740 | 0.478 | 7.317 * | 6.819 * | 0.000 |
| NRE does not Granger cause UR | 5.659 * | 0.004 | 4.851 * | 3.574 * | 0.000 |
| UR does not Granger cause NRE | 4.764 * | 0.009 | 8.113 * | 7.866 * | 0.000 |
| RE does not Granger cause UR | 2.010 | 0.135 | 9.311 * | 9.442 * | 0.000 |
| UR does not Granger cause RE | 0.782 | 0.45 | 4.604 * | 3.249 * | 0.001 |
| TO does not Granger cause FD | 0.936 | 0.39 | 4.084 ** | 2.566 ** | 0.010 |
| FD does not Granger cause TO | 1.035 | 0.356 | 4.094 * | 2.579 * | 0.009 |
| FD does not Granger cause NRE | 1.208 | 0.299 | 3.675 ** | 2.029 ** | 0.042 |
| RE does not Granger cause FD | 0.197 | 0.821 | 3.965 ** | 2.410 ** | 0.016 |
| NRE does not Granger cause TO | 2.514 *** | 0.082 | 4.934 * | 3.685 * | 0.000 |
| RE does not Granger cause TO | 2.346 *** | 0.097 | 5.006 * | 3.779 * | 0.000 |

Note: the */**/*** represent 1%, 5%, and 10% significance levels, respectively. The lag length specified is 2.

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
