# Peer review of "The Influence of Industrial Output, Financial Development, and Renewable and Non-Renewable Energy on Environmental Degradation in Newly Industrialized Countries"

_sustainability, doi:10.3390/su15064742_

Round 1
Reviewer 1 Report
The study identifies key issues regarding environmental factors for the sustainable development and investment goals. Some specific issues should be improved.
1. The abstract part should be improved. The main reason of selecting this topic should be given at the beginning of this part.
2. More strategies should be given at the end of the abstract part based on the analysis results.
3. The contributions of this study should be explained at the end of the introduction part in a more detailed manner.
4. I suggest minor English language editing and avoiding long complex sentences that go apart from main idea.
5. The contribution of the manuscript should be explained more in the conclusion.
6. More information should be given about the limitations and future Research directions in the conclusion part.
7.Just after introducing variables authors pass to the hypothesis, however, it would be better to add a short information about data, its range, descriptives and others.
8.Results should be followed with a discussions part to bring all hypothesis together and conclude. You should avoid using variable acronyms in this part, in this case reader needs to go back previuos part to see what that block letter acronym stand for.
9.After minor revisions, the manuscript is acceptable.
Author Response
Esteemed Reviewer 1
Thank you for providing us with the chance to improve our paper, and thank you for your comments and suggestions, which have helped us improve our paper and make it flow better. We have amended the paper as requested, which can be seen in the track changes, and the answers provided in the second column next to your comments and suggestions in the first column.
|
Reviewer #1 comments |
Remarks |
|
1. The abstract part should be improved. The main reason for selecting this topic should be given at the beginning of this part. |
As suggested, we added the study's objective at the start of the abstract. |
|
2. More strategies should be given at the end of the abstract part based on the analysis results. |
We have added this as suggested—Vide track changes. |
|
3. The contributions of this study should be explained at the end of the introduction part in a more detailed manner. |
We have revised our contribution part. Please see page 3, Lines 110-12. |
|
4. I suggest minor English language editing and avoiding long complex sentences that go apart from main idea. |
Necessary editing and language errors have been rectified by one of the co-authors, who is a native English speaker with a high level of academic knowledge of English grammar. |
|
5. The contribution of the manuscript should be explained more in the conclusion |
As suggested, we have added the study's contribution to the conclusion. Please see page 19, Lines 580 to 588. |
|
6. More information should be given about the limitations and future Research directions in the conclusion part. |
Future research directions and limitations have been added. Please see page 19, Lines 622 to 630. |
|
7. Just after introducing variables authors pass to the hypothesis, however, it would be better to add a short information about data, its range, descriptives and others. |
Variables' description and data source is provided in table 4, while descriptive statistics and correlation matrix are discussed in section 4 (see table 5). Pg. 8 & 13-14 |
|
8. Results should be followed with a discussions part to bring all hypothesis together and conclude. You should avoid using variable acronyms in this part, in this case reader needs to go back previous part to see what that block letter acronym stands for. |
We have addressed the valid comment and all the acronyms were explained (Pg. 17-18) |
Reviewer 2 Report
Referee report on manuscript Sustainability-2205934
The influence of industrial output, financial development, renewable and non-renewable energy on environmental degradation in Newly Industrialized Countries
The manuscript "The Influence of Industrial Output, Financial Development, Renewable and Non-Renewable Energy on Environmental Degradation in Newly Industrialized Countries" investigates the effects of industrial output, financial development, and renewable and non-renewable energy on CO2 emissions in NICs. The manuscript contains significant drawbacks that should be addressed before being accepted for publication in Sustainability.
1. The authors should elaborate on the relevance of their research into the factors affecting environmental sustainability in NICs. Why is NICs a good case study?
2. It is preferred to discuss stylized facts on the industrialization process in NICs.
3. How did the authors select the variables introduced in the specification?
4. In the STIRPAT model, the dependent variable should be CO2 emissions and not CO2 emissions per capita. Please refer to the previous literature, modify the dependent variable, and re-estimate the model.
5. Please add descriptive statistics and the correlation matrix.
6. In the presence of cross-section dependence, Pedroni and Kao cointegration tests are no longer valid. In this case, I recommend performing the cointegration test developed by Westerlund (2007).
7. I recommend using the PMG-ARDL model to estimate the short- and long-run effects. The FMOLS is not suitable for this study as it does not allow estimating the short-run effects and does not account for cross-section dependence in the residuals.
8. Authors could also test the presence of cross-section dependence in residuals obtained from the PMG-ARDL model.
9. The abstract needs rewriting. Only significant findings must be provided. For example, the results of unit root tests may not be included in the abstract.
Author Response
Esteemed Reviewer 2
Thank you for providing us with the chance to improve our paper, and thank you for your comments and suggestions, which have helped us improve our paper and make it flow better. We have amended the paper as requested, which can be seen in the track changes, and the answers provided in the second column next to your comments and suggestions in the first column.
|
Reviewer #2 comments |
Remarks |
|
1. The authors should elaborate on the relevance of their research into the factors affecting environmental sustainability in NICs. Why is NICs a good case study? |
We have considered the NICs panel for the following reasons; 1) The newly industrialised countries (NIC) have been more responsible for CO2 emissions since 1990 than the industrialised ones. CO2 is a part of the atmosphere but has increased dramatically by one-third after the industrial revolution NASA data (2016) shows that atmospheric CO2 remained below 300 parts per million (ppm) for 650,000 years but is now at 400 ppm. It depicted that CO2 emissions grew with the industrial revolution. More energy use in industries is a key factor behind high CO2 emissions. More energy use in sectors of the economy could be a major factor behind high CO2 emissions, although it's not a single factor. Please see page 2, lines 64 to 71. 2) This study analysed the impact of industrial output and financial development in NICs. To the best of our knowledge, in the case of NICs, only a few studies have examined the causal nexus between carbon emissions and variables like GDP, energy consumption, trade openness, and urbanisation. However, industrial output and financial development have not garnered much attention from researchers. |
|
2. It is preferred to discuss stylised facts on the industrialisation process in NICs |
We did not detail stylised facts on the industrialisation process in NICs as the length of the paper does not allow the creation of separate sections. However, a plausible reason for the consideration of NICs has been provided in the introduction section. Please see page 2, lines 64 to 71. |
|
3. How did the authors select the variables introduced in the specification? |
We have addressed the suggested comment on Pg. 8 of the article. |
|
4. In the STIRPAT model, the dependent variable should be CO2 emissions and not CO2 emissions per capita. Please refer to the previous literature, modify the dependent variable, and re-estimate the Model.
|
Following other quality literature, we used the dependent variable as CO2 emissions per capita. The relevant studies are cited for reference (Pg. 8) |
|
5. Please add descriptive statistics and the correlation matrix. |
The descriptive statistics have already been done. However, we have added the correlation matrix and discussed it. (Pg. 13) |
|
6. I recommend using the PMG-ARDL Model to estimate the short- and long-run effects. The FMOLS is not suitable for this study as it does not allow estimating the short-run effects and does not account for cross-section dependence in the residuals. |
We agree with the reviewer's comment that ARDL is suitable for both the estimated long-run and short-run effects. In the current study, we are interested in estimating variables' long-run effects and causality. In the available literature, different studies used the method (Pg. 17) |
|
7. Authors could also test the presence of cross-section dependence in residuals obtained from the PMG-ARDL model. |
Though we did not address the ARDL, the residuals obtained from FMOLS are checked for stationarity, ensuring that the Model's residuals are white and our results are not spurious (Pg. 17). |
|
8. The abstract needs rewriting. Only significant findings must be provided. For example, the results of unit root tests may not be included in the abstract. |
We have revised the abstract and excluded unit root test results as suggested. |
Round 2
Reviewer 2 Report
The manuscript has made progress but still contains some drawbacks that should be addressed.
1. The authors tried to explain the motivations for choosing NICs by adding a paragraph to the introduction. However, I think they failed to do it because they do not provide sources when indicating that “The newly industrialised countries (NIC) have been more responsible for CO2 emissions since 1990 than the industrialised ones”. They should add some sources, such as statistics and reports/articles, confirming this assertion. Moreover, the next sentence deals with global atmospheric CO2 emissions, not NICs, which we think is inappropriate. Additional work on this issue is needed.
2. In their response to my comments, the authors affirm that they are interested in estimating the long-run effects, but in conclusion, they wrote, “This study attempts to find long-run and short-run associations between industrial output, financial development, gross domestic product…”! Why study only the long-run effects? This should be justified.
3. What are the exact sources of variables “renewable energy consumption” and “non-renewable energy consumption”? The authors added two sources for each variable!
4. The article contains a lot of typos and grammatical errors. For example, “The Bruesch-Pagen LM” in the abstract should be “The Bruesch-Pagan LM”. The whole manuscript should be edited by a native English speaker.
5. At the end of the conclusion, the authors wrote, “Future research on the theme should focus on demographic variables such as democracy and population”. Democracy is not a demographic factor. Please make a careful read of the manuscript.
6. There should be no abbreviations in the abstract because they may cause readers to become confused.
Author Response
Dear Editor,
We would like to thank you and the esteemed reviewers for their comments and suggestions on our article. We have revised our article in line with the recommendations made by the reviewers. The main changes are highlighted in blue in the text and the rest are in track changes. As an answer to the comments and suggestion for ease of the reviewers we have answered in bold in the covering letter below the comment. We hope that all these changes fulfill your requirements as well as those of the reviewers. Many thanks for providing us with this opportunity for improving our article.
Reviewers' comments:
Comment#1: The authors tried to explain the motivations for choosing NICs by adding a paragraph to the introduction. However, I think they failed to do it because they do not provide sources when indicating that “The newly industrialised countries (NIC) have been more responsible for CO2 emissions since 1990 than the industrialised ones”. They should add some sources, such as statistics and reports/articles, confirming this assertion. Moreover, the next sentence deals with global atmospheric CO2 emissions, not NICs, which we think is inappropriate. Additional work on this issue is needed.
- Authors response: Thanks for highlighting this issue. We have removed the particular sentence mentioned in the comment as we were not able to find an appropriate citation for it. We have added more text with proper references to justify that NICs are increasing the environmental degradation. Please see page 2, Lines 62 to 70.
Comment#2: In their response to my comments, the authors affirm that they are interested in estimating the long-run effects, but in conclusion, they wrote, “This study attempts to find long-run and short-run associations between industrial output, financial development, gross domestic product…”! Why study only the long-run effects? This should be justified.
- Authors response: We made the correction highlighted in the conclusion, and added the following:
“the main objective is to find the long-run estimates of CO2 emissions and the causal relationship between industrial output, financial development, gross domestic product.
Furthermore, we added clarifications in the methodology section. However, we have focused primarily on long run analysis with some justifications.
We have used variables such as financial development industrial output as well as the outcome variable since they have a profound impact in the long run compared to short run and moreover long run effect also assist in providing a comprehensive picture as compared to the short run. Therefore, the long run effects are general more useful for policy constructions. Also, we agree with the reviewer and will be considering short run effects as well.
Comment#3: What are the exact sources of variables “renewable energy consumption” and “non-renewable energy consumption”? The authors added two sources for each variable!
- Authors response: We incorporated and addressed these clarification on page-9.
Comment#4: The article contains a lot of typos and grammatical errors. For example, “The Bruesch-Pagan LM” in the abstract should be “The Bruesch-Pagan LM”. The whole manuscript should be edited by a native English speaker.
- Authors response: The article has been reviewed by one of the authors who is a native English speaker.
Comment#5: At the end of the conclusion, the authors wrote, “Future research on the theme should focus on demographic variables such as democracy and population”. Democracy is not a demographic factor. Please make a careful read of the manuscript
Authors’ response: Thanks for highlighting mistake. We have corrected it.
Comment#6: There should be no abbreviations in the abstract because they may cause readers to become confused.
- Authors response: We have addressed this as per suggestions made.
Once again, thank you very much. We appreciate the reviewers’ comments, and hope that the correction will meet the approval
Sincerely
Prof. Simon Grima
